# Solving Urban Network Security Games: Learning Platform, Benchmark, and Challenge for AI Research

## Abstract

After the great achievement of solving two-player zero-sum games, more and more AI researchers focus on solving multiplayer games. To facilitate the development of designing efficient learning algorithms for solving multiplayer games, we propose a multiplayer game platform for solving Urban Network Security Games (**UNSG**) that model real-world scenarios. That is, preventing criminal activity is a highly significant responsibility assigned to police officers in cities, and police officers have to allocate their limited security resources to interdict the escaping criminal when a crime takes place in a city. This interaction between multiple police officers and the escaping criminal can be modeled as a UNSG. The variants of UNSGs can model different real-world settings, e.g., whether real-time information is available or not, and whether police officers can communicate or not. The main challenges of solving this game include the large size of the game and the co-existence of cooperation and competition. While previous efforts have been made to tackle UNSGs, they have been hampered by performance and scalability issues. Therefore, we propose an open-source UNSG platform (**GraphChase**) for designing efficient learning algorithms for solving UNSGs. Specifically, GraphChase offers a unified and flexible game environment for modeling various variants of UNSGs, supporting the development, testing, and benchmarking of algorithms. We believe that GraphChase not only facilitates the development of efficient algorithms for solving real-world problems but also paves the way for significant advancements in algorithmic development for solving general multiplayer games.

## 1 Introduction

In the field of AI research, a lot of focus has been placed on computing a Nash equilibrium (Nash, 1951; Shoham & Leyton-Brown, 2008) in two-player zero-sum extensive-form games, where both players receive opposing payoffs (Zinkevich et al., 2008; Moravčík et al., 2017; Brown & Sandholm, 2018). In this scenario, a Nash equilibrium can be computed in polynomial time based on the size of the extensive-form game (Shoham & Leyton-Brown, 2008). Recent significant achievements, such as achieving superhuman performance in the heads-up no-limit Texas hold'em poker game (Moravčík et al., 2017; Brown & Sandholm, 2018), demonstrate that researchers have a good understanding of the problem of computing a Nash equilibrium in two-player zero-sum extensive-form games, both in theory and in practice. However, the problem of computing a Nash equilibrium in multiplayer games is not as well understood, as it is generally a challenging task (Chen & Deng, 2005; Zhang et al., 2023b). Therefore, more and more AI researchers focus on solving multiplayer games (Brown & Sandholm, 2019; FAIR et al., 2022; Carminati et al., 2022; Zhang et al., 2023a; McAleer et al., 2023; Zhang et al., 2024)

To facilitate the development of designing efficient learning algorithms for solving multiplayer games, we propose a multiplayer game platform for solving Urban Network Security Games (**UNSGs**) that model the following real-world situations. In urban areas, ensuring public safety and security is crucial for law enforcement agencies. One significant challenge they face is the high number of innocent bystanders who are injured or killed during police pursuits (Rivara & Mack, 2004). It's essential to develop effective strategies that allow multiple officers to apprehend fleeing criminals while minimizing risks to civilians and property damage. This paper focuses on respond-

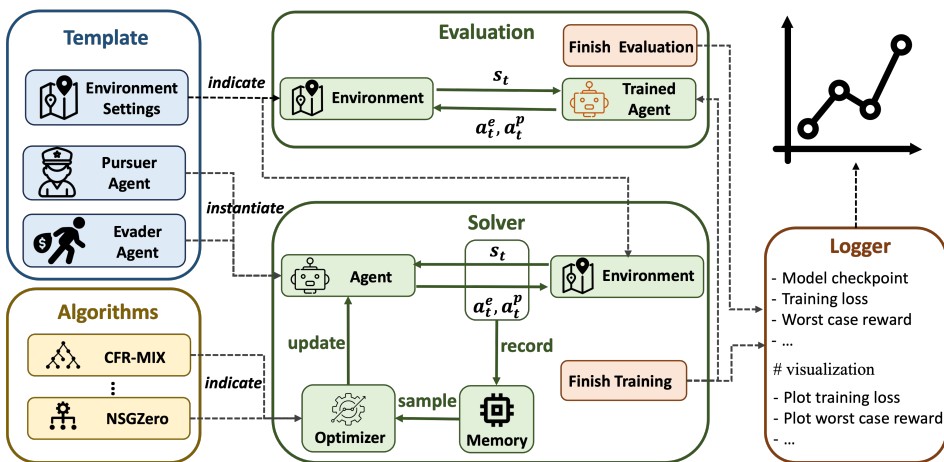

Figure 1: The blueprint of our GraphChase platform.

ing to major incidents such as terrorist attacks or bank robberies, where police officers need to swiftly intercept the attackers during their escape. This requires efficient strategies for apprehending fleeing criminals, which can be analyzed and developed using structured approaches like UNSGs.

However, solving UNSGs is NP-hard (Jain et al., 2011; Zhang et al., 2017; 2019). More specifically, the strategy space of players in UNSGs cannot be enumerated due to the memory constraint of computers (Jain et al., 2011; Zhang et al., 2019). Moreover, if players do not have real-time information, they have to make decisions with imperfect information. In addition, if police officers cannot communicate during the game play, they have to make decisions independently. Finally, UNSGs incorporate cooperation between police officers and competition between the criminal and team of police officers. To address the above challenges, previous efforts have been made to tackle UNSGs. That is, they extended the Counterfactual Regret Minimization (CFR) (Zinkevich et al., 2008) to CFR-MIX algorithm (Li et al., 2021), incorporating deep learning enhancements from Deep CFR (Brown et al., 2019). Additionally, they utilized the Neural Fictitious Self-Play (NFSP) approach (Heinrich & Silver, 2016), further developed into NSG-NFSP (Xue et al., 2021) and NSGZero (Xue et al., 2022), which are tailored to solving UNSGs under the NFSP framework. Moreover, they extended the learning framework, Policy-Space Response Oracles (PSRO) (Lanctot et al., 2017), to an advanced variant Pretrained PSRO (Li et al., 2023a) to speed up. Finally, they developed Grasper (Li et al., 2024) based on Pretrained PSRO, an innovative algorithm that can generalize across different game settings. All of them are based on the state-of-the-art game-theoretical algorithm frameworks. However, these efforts are still hampered by performance and scalability issues, as shown in our experiments.

To foster the development of scalable algorithms capable of addressing city-scale UNSGs, we propose the creation of an open-source platform, **GraphChase**, specifically tailored for UNSG. The architecture of GraphChase is depicted in Figure 1, designed to provide researchers with a comprehensive UNSG platform and facilitate the development and evaluation of scalable strategy for pursuers. Specifically, we made the following contributions: i) **Development of a unified and flexible UNSG environment:** We developed a versatile platform designed to support various configurations of UNSGs. Specifically, this environment allows for modifying game parameters, enabling researchers to simulate different real-world UNSG scenarios under various conditions. The inherent flexibility of GraphChase supports a wide range of experimental setups, from small-scale laboratory experiments to city-wide simulations. All these make GraphChase a suitable tool for theoretical research and practical application testing. ii) **Implementation of learning algorithms:** GraphChase is designed to facilitate the execution of a wide range of algorithms. Based on the standardized platform, we successfully implement several advanced deep learning-based algorithms, enabling the consistent comparison of different strategic approaches. By efficiently integrating algorithms within the platform, it reduces the time overhead of the simulation resulting in faster convergence from the perspective of wall-clock time. And iii) **Benchmark results:** We conduct experiments on UNSGs with synthetic and real-world graphs to evaluate the performance of the different algorithms

implemented on the GraphChase platform. The results from these experiments are recorded and compiled into comprehensive benchmarks. Our results show that, although previous algorithms can achieve reasonable performance, they still suffer performance and scalability issues in real-world settings. These results suggest that substantial efforts are still required to develop effective and efficient algorithms for solving real-world UNSGs. We believe that GraphChase not only facilitates the development of efficient algorithms for solving real-world problems but also paves the way for significant advancements in algorithmic development for solving general multiplayer games.

## 2 URBAN NETWORK SECURITY GAMES

Motivated by the security games on urban roads (Jain et al., 2011; Zhang et al., 2017; 2019), we proposed our GraphChase platform for solving UNSGs that model the interactions between multiple pursuers (police officers) and an evader (criminal). The variants of UNSGs can model different real-world settings, e.g., whether real-time information is available, whether pursuers can communicate. Now, we introduce the definition of these games.

### 2.1 GAME DEFINITION

Take, for instance, the scenario where pursuers are tasked with capturing an evader escaping on urban roads. We introduce the concept of UNSGs. First, urban roads and pathways naturally lend themselves to being modeled as graphs, where intersections and streets form nodes and edges, respectively. The graph can be represented by $G = (V, E)$, where $V$ is a set of vertices, and $E$ is a set of edges. In UNSGs, graphs can be directed or undirected, corresponding to one-way streets and two-way streets, and weighted or unweighted, where the weight can be used to reflect different travel costs or terrains. This graphical representation allows for a structured and systematic approach to simulating the complex dynamics of urban pursuits. Specifically, in graph $G$, we use a subset of the vertex set, $E_{exit} \subset E$, to represent the set of exit nodes from which the criminal can escape. For each vertex $v \in V$, we use $\mathcal{N}(v)$ to represent the set of neighbours of $v$.

In UNSGs, the pursuer and the evader are represented as agents moving across a network. It is important to note that the evader and the pursuer can be a single agent or consist of multiple agents. For example, several pursuers would collaborate to chase a single evader or chase a team of evaders. Formally, the set of players $N = (\mathbf{p}, \mathbf{e})$, where $\mathbf{p} = (p_1, p_2, ..., p_n), n \geq 1$ represents pursuers and $\mathbf{e} = (e_1, e_2, ..., e_n), n \geq 1$ represents the evader. Since the pursuers can block all exit nodes for a certain period, we can predefine the length of the lockdown. Formally, let $T$ represent the number of steps in which the game terminates and $l_0^{\mathbf{p}} = (l_0^{p_1}, l_0^{p_2}, ..., l_0^{p_n})$, $l_0^{\mathbf{e}} = (l_0^{e_1}, l_0^{e_2}, ..., l_0^{e_n})$ represent the initial locations of the evader and the pursuer, respectively. At each step, each player in the game would move from vertex $v$ to one of its neighborhood vertices $w \in \mathcal{N}(v)$. Specifically, at game step $t < T$, the locations of the evader and the pursuer, respectively, are $l_t^{\mathbf{p}} = (l_t^{p_1}, l_t^{p_2}, ..., l_t^{p_n})$, $l_t^{\mathbf{e}} = (l_t^{e_1}, l_t^{e_2}, ..., l_t^{e_n})$. Then the available action set of the pursuer is a Cartesian product of the sets of neighboring vertices of each evader, i.e., $A_{\mathbf{p}}(h) = \{(l^{p_1}, l^{p_2}, ..., l^{p_n}) | l^i \in \mathcal{N}(l_t^i), \forall i \in \{p_1, p_2, ..., p_n\}\}$. Similarly, the available action set of the evader is $A_{\mathbf{e}}(h) = \{(l^{e_1}, l^{e_2}, ..., l^{e_n}) | l^i \in \mathcal{N}(l_t^i), \forall i \in \{e_1, e_2, ..., e_n\}\}$. All players act simultaneously at game step $t$, i.e., the pursuer and the evader select actions from their action sets. Then all players move from $l_t^{\mathbf{p}}$ and $l_t^{\mathbf{e}}$ to $l_{t+1}^{\mathbf{p}}$ and $l_{t+1}^{\mathbf{e}}$, respectively. We can also convert the simultaneous-move game into a turn-based game by ignoring the selected action of the first-act player when the second player acts. This process repeats until a termination condition is met. The evader is considered caught if the evader and any of the pursuers occupy the same point at any time within the maximum time horizon. The termination conditions of the game include: (i) the pursuer catches the evader (i.e., all criminals); (ii) the evader (i.e., all criminals) escapes from exit nodes; and (iii) the game reaches the predefined game step $T$, i.e., $t = T$. In cases (i) and (iii) the pursuer wins. Otherwise, if the evader successfully escapes to the outside world, the evader wins. Based on these results, each player gets their corresponding rewards.

### 2.2 INFORMATION AND STRATEGY

In different real-world cases, the pursuer and evader may access various information, i.e., the location information of each player. With the aid of tracking devices, such as the StarChase GPS-based system (Gaither et al., 2017), police officers can get the real-time location of the criminal. In another

case, the police officers may not know the ability of the criminal. To avoid the worst case, the police officers usually assume that the criminal can get the real-time location of the police officers. Therefore, there are four cases: i) the evader can get the real-time location information of the pursuer while the pursuer cannot get the real-time location information of the evader; ii) the pursuer can get the real-time location information of the evader while the evader cannot get the real-time location information of the pursuer; iii) both the evader and the pursuer can get the real-time location information of the opponent; and iv) both the evader and the pursuer cannot get the real-time location information.

Moreover, if pursuers cannot communicate during the game play, they have to make decisions independently. However, if pursuers can communicate during the game play, they can correlate their actions. Using this case as an example, based on the available real-time location information, the behavior strategy $\sigma_\mathbf{e}$ or $\sigma_\mathbf{p}$ is a function that maps every decision point to a probability distribution over the available action set. Then, a strategy profile $\sigma$ is a tuple of one strategy for each player, i.e., $\sigma = (\sigma_\mathbf{p}, \sigma_\mathbf{e})$. The pursuer's payoff function is $u_\mathbf{p}(\sigma_\mathbf{p}, \sigma_\mathbf{e}) \in \mathbb{R}$ with $u_\mathbf{p}(\sigma_\mathbf{p}, \sigma_\mathbf{e}) = -u_\mathbf{e}(\sigma_\mathbf{p}, \sigma_\mathbf{e})$ for the evader. We adopt the Nash equilibrium (NE) (Nash, 1950) as the solution concept for this case since the NE strategy profile is a steady state in which no player can increase its utility by unilaterally deviating. In our GraphChase platform, we consider the NE strategy of the pursuer would be the optimal strategy and take the worst-case utility of the pursuer as the measure for the pursuer's strategy, i.e., $\max_{\sigma_\mathbf{p} \in \Sigma_\mathbf{p}} \min_{\sigma_\mathbf{e} \in \Sigma_\mathbf{e}} u_\mathbf{p}(\sigma_\mathbf{p}, \sigma_\mathbf{e})$.

### 2.3 CHALLENGES

In UNSGs, pursuers are tasked with capturing an evader escaping on urban roads. The network-based environment could lead to the strategy space of players in UNSGs cannot be enumerated due to the memory constraint of computers (Jain et al., 2011; Zhang et al., 2019). That is, if a player's strategy is a path, then we cannot enumerate all paths due to memory constraints in large-scale UNSGs. In fact, even with the relatively simple setting where the time dynamics are ignored, and the pursuers can correlate their actions, the problem of solving UNSGs is still very hard (Jain et al., 2011). We could expect that solving UNSGs will be harder in more complicated settings.

Moreover, some UNSGs operate under conditions of imperfect information when real-time information is not available. In some cases, players possess asymmetric knowledge about the state of the environment. In some UNSGs, the escaping evader location and potential strategies might not be fully known to the pursuers in some scenarios, and conversely, the evader may have limited information about the evader locations. The partial observability also poses unique challenges for addressing the UNSGs. In some cases, the maximum number of time steps may not be predicted accurately. Therefore, it necessitates the development of robust algorithms capable of making decisions based on imperfect data and under uncertainty, requiring sophisticated decision-making processes akin to those used in real-world scenarios.

Furthermore, pursuers cannot communicate during the game play in some UNSGs, and then they have to make decisions independently. This case is similar to general multiplayer games, where NE is commonly used as a solution concept. However, computing an NE is hard generally (Chen & Deng, 2005; Zhang et al., 2023b).

In addition, the UNSG, inherently a zero-sum game, involves direct competition between the pursuers and the escaping evader, where one's gain is precisely the other's loss, reflecting the purely adversarial nature of their interactions. Concurrently, profound cooperation within the team of pursuers is also essential. pursuers must work together seamlessly to effectively capture the escaping evader. The pursuers share the same utility function, aiming collectively to minimize the escape possibilities of the evader. This blend of competitive and cooperative elements introduces significant complexities in solving UNSGs. The dual nature of interactions demands algorithms that can handle both aspects simultaneously—optimizing competitive moves against the escaping evader while coordinating strategies among multiple pursuers.

These elements—combined competitive and cooperative dynamics, along with the challenge of operating under imperfect information or independent moves — make the UNSG an exemplary benchmark for assessing the effectiveness of algorithms in complex and unpredictable environments. By providing a platform that mimics the diverse scales and complexities of UNSGs, GraphChase offers a valuable tool for advancing the development of scalable algorithms.

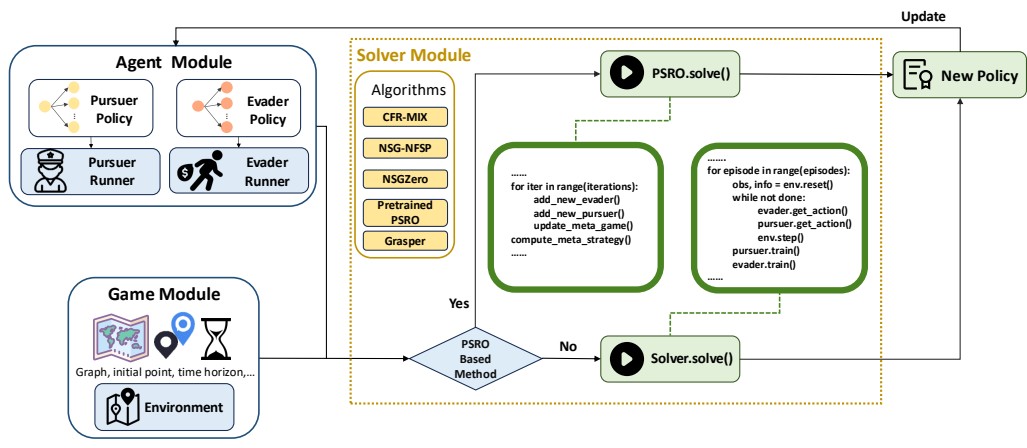

Figure 2: The core structure and workflow of GraphChase.

# 3 PLATFORM: GRAPHCHASE

As shown in Figure 1, GraphChase provides template scripts for quick-start, and, once completed by the user, it carries out training and testing procedures for comparison. Results, such as the worst-case reward, are generated and available for review.

## 3.1 CORE COMPONENTS

Our GraphChase platform features a flexible game environment specifically designed to facilitate comprehensive simulations of UNSGs. The parameters that users can control to generate the graph structure are detailed in Appendix C. There is a brief introduction about how to use GraphChase in Appendix F. At the core of this environment is a versatile system architecture, as depicted in Figure 2, which clearly outlines the primary components and their interactions within the platform. The modular architecture comprising the **Game Environment**, **Agent**, and **Solver** components enhances platform versatility, facilitating both adaptation to diverse research requirements and integration of various algorithmic approaches. This modular design architecture enables researchers to easily customization and scale their own problems.

**Game Module.** To enhance the flexibility of our platform, GraphChase is designed to support extensive customization of game parameters, enabling users to simulate different UNSG scenarios tailored to their specific demands. This customization capability includes several key features. First, users have the option to design or import their graphs for simulation. This could range from simple, manually-generated grid diagrams to more complex real-world urban layouts, such as the Singapore road map. Any graph format can be transformed into an adjacency list as the input to the game generation function. This feature allows researchers to explore UNSG in simulations that are directly relevant to their specific areas of study or practical application needs. Second, users can specify key strategic points within the graph, such as initial positions of the pursuer and the evader, and exit nodes. This level of customization not only adds complexity and variability to the simulations but also allows for testing strategies under different initial conditions and escape routes, making each game unique even when played on the same graph. Third, the platform supports customization of the time horizon for each game, accommodating both quick resolutions and longer strategic engagements. Fourth, since GraphChase is based on the Gymnasium library, the amount and type of information accessible to each player can be easily adjusted by users via the API of *gymnasium.Env.step()*. This feature allows the evader and the pursuer to have limited visibility of each other's locations and moves, creating more realistic scenarios that closely replicate the information asymmetry often found in real-world situations. In conclusion, by allowing users to freely define the structure of the graph, GraphChase enables a broad spectrum of simulation possibilities. The flexibility of GraphChase allows users to meticulously design games that meet their specific research or operational requirements. Furthermore, through integration with the Gymnasium library, users can significantly reduce the time to learn and utilize GraphChase, while also leveraging various Gym-

nasium wrappers to conveniently run environments in parallel and visualize the performance of the trained models.

**Agent Module.** The Agent Module consists of two parts: the agent policy and the agent runner. The policy refers to the algorithms adopted by the agent, such as PPO, MAPPO, and NSGZero. The agent runner is responsible for simulation in the environment against opponents and uses the obtained data to update the agent policy. Specifically, an agent runner must have a *get_action(data)* method, where data is a tuple providing the input required for the agent policy to generate actions. The actions made by the policy are returned as the output of the *get_action()* method. Additionally, if the agent needs to improve its policy (not necessary in some cases, such as random strategies), it must have a *train()* method. Users can freely define this method according to the requirements of their designed algorithms. In summary, with this agent module structure, users can customize pursuers and evaders adopting various algorithms and can easily integrate with the Game module introduced earlier and the solver module discussed later in the paper. This flexible structure provides a rich testing ground for developing both defensive and offensive strategies within the game environment, allowing users to test the performance of different algorithms efficiently.

**Solver Module.** The Solver module of our GraphChase platform encompasses a variety of algorithms designed to address UNSGs, aiming to facilitate users in comparing the performance of various algorithms. Given that the current state-of-the-art algorithms, such as Pretrained PSRO and Grasper, are based on the PSRO framework, we have integrated the PSRO learning framework within our platform to solve UNSGs. Users merely need to define the training methods for both the pursuer runner and the evader runner and provide the environment with parameters to initialize the PSRO algorithm. By designing the code structure in this manner, users can freely modify the algorithms used by the pursuer, such as PPO or MAPPO, and seamlessly integrate them within the PSRO framework, thereby maximizing code reusability. Additionally, if users design a new learning framework and want to compare its performance to PSRO, they only need to define the environment and agents as introduced before, then a training task can be easily started.

### 3.2 BENCHMARK ALGORITHMS

Based on our GraphChase platform, we have implemented several deep-learning algorithms that solve UNSGs. Here, we provide a brief overview of these algorithms and outline their operational process within our platform, as illustrated in Figure 2.

To address the inherent challenges of imperfect information in UNSGs, we integrate several sophisticated algorithms into GraphChase. It includes: 1) **CFR-MIX** algorithm (Li et al., 2021), incorporating deep learning enhancements (Brown et al., 2019) based on counterfactual regret minimization (CFR) (Zinkevich et al., 2008); 2) **NSG-NFSP** (Xue et al., 2021) based on the neural fictitious self-play approach (Heinrich & Silver, 2016); 3) **NSGZero** (Xue et al., 2022) based on neural Monte Carlo tree search; 4) Variants of the PSRO framework (Lanctot et al., 2017): **Pretrained PSRO** (Li et al., 2023a) and **Grasper** (Li et al., 2024). Figure 2 illustrates the operational process of these algorithms within our GraphChase platform.

Each algorithm is implemented to integrate with the game's underlying mechanics through well-defined interfaces, ensuring they can operate effectively within the platform's architecture. Firstly, by inputting the initial positions of agents and the time horizon of the game, we set up the game environment. Simultaneously, we initialize the pursuer runner according to the solver algorithms, as shown in the yellow frame. Then, depending on whether the chosen algorithm requires the PSRO learning framework, different solving processes are employed. If the PSRO framework is not required, the *solver.solve()* method is executed. In this method, the evader and pursuer runner interact continuously with the environment to generate data, which is then used to update the policy network, producing new strategies. If the PSRO framework is used, the *PSRO.solve()* method is executed. During each iteration, the opponent's strategy is alternately fixed, and a best response to the opponent is generated. The meta game is then updated based on the simulation results, and the current policy oracle's meta strategy is computed. Subsequently, the runner's policies are updated, and the process proceeds to the next training cycle.

**Evaluation.** In our platform, the primary objective is to compute the optimal defense strategy for the pursuer, akin to strategizing the most effective tactics for police officers in realistic scenarios. Upon determining the pursuer's strategy through any of the algorithms available on the platform,

we adopt the worst-case utility as our principal evaluation metric. As introduced before, to compute the worst-case utility, we first identify the best response strategy of the evader against the pursuer's strategy being evaluated. Then, we compute the pursuer's worst-case utility by simulating the game using the pursuer's strategy and the best response strategy of the evader. This evaluation method helps ensure that the strategy is not only theoretically sound but also practically viable under the most demanding conditions.

## 4 EXPERIMENTAL EVALUATION

We conduct experiments to evaluate GraphChase and show the issues of existing algorithms.

### 4.1 EXPERIMENTAL SETTING

We conduct the following three sets of experiments for the experimental evaluation. 1) To evaluate the effectiveness of GraphChase, we compare the training procedure of algorithms implemented in GraphChase with the training procedure of algorithms implemented by the original authors[1]. 2) To evaluate the performance of existing algorithms, we calculate the reward (the probability of catching the evader) of the pursuer in the worst-case setting. That is, the pursuer's policy in a trained model will be played against all available paths of the evader, and the worst-case reward will be the reward of this model. 3) To evaluate the scalability of existing algorithms, we run these algorithms to solve realistic and large games.

We run the first two sets of experiments on the following two games shown in Appendix A. The first game is easier to solve as the evader will be caught with a probability of 1 (ground truth), but the second game is harder to solve as the evader will only be caught with a probability of 0.5 (ground truth). Both games are run on a $7 \times 7$ grid network with four exits, four pursuers, and one evader. In the first set of experiments, we set $T = 6$. In the second set of experiments, we evaluate the pursuers' trained model in the first set of experiments against all paths of the evader with the maximum length of each path as 6 and 12, respectively. Finally, we conducted a third set of experiments on a game set in a $100 \times 100$ grid network with a maximum time horizon of $T = 200$. In this scenario, four pursuers attempt to capture a single evader who is trying to escape successfully through one of 12 exit nodes.

### 4.2 BENCHMARK RESULTS

**The Effectiveness of GraphChase**. The results of the evaluation of GraphChase are shown in Figures 3 and 4 (Results for other algorithms are in Appendix B). We can see that the algorithms based on our GraphChase perform similarly to the algorithms based on the original codes. In most cases, we can see that algorithms based on GraphChase converge faster than the algorithms based on the original codes, which shows the effectiveness and efficiency of our GraphChase.[2]

To further verify that algorithms based on GraphChase can recover the performance of the algorithms based on the original codes with significantly less time, we first show that our algorithms based on GraphChase can recover the performance of the algorithms based on the original codes in a variety of scenarios used in the UNSG domain (Xue et al., 2021; 2022; Li et al., 2023a; 2024) in Appendix D. These networks, including the $15 \times 15$ grid network, the real-world Singapore map, and the real-world Manhattan map, are representatives because the $15 \times 15$ grid network represents the randomly generated network, and two real-world networks represent different topological structures in real-world cities. Then, in Appendix E, we show that algorithms based on GraphChase run significantly faster than algorithms based on the original codes in terms of simulation and data-saving time, and we explain the reasons behind the faster convergence of GraphChase.

**The Performance Issue of Existing Algorithms**. The performance evaluation of existing algorithms for solving UNSGs is shown in Table 1. We can see that if an algorithm converges during training, it will perform well for solving the easy game (with a caught probability of 1) but may not perform well for solving the hard game (with a caught probability of 0.5). Increasing the maximum length of the evader's paths also will damage the performance.

---

[1]Codes were shared by the original authors of these algorithms.
[2]CFR-MIX and NSGZero solved games on $5 \times 5$ network with $T = 4$ because they run too slow.

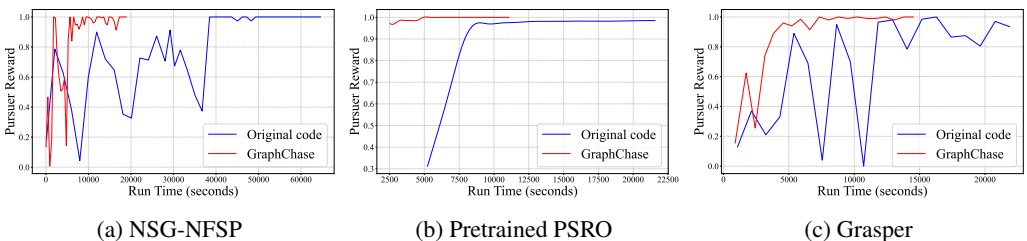

Figure 3: The training procedure on the easy game with a caught probability of 1.

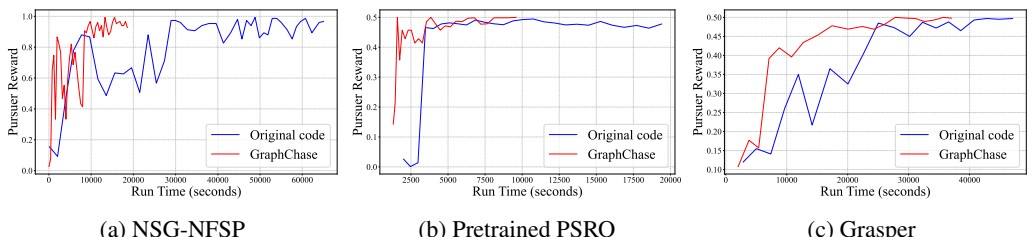

Figure 4: The training procedure on the hard game with a caught probability of 0.5.

The main reason is that, when the evader does not have real-time location information of pursuers, computing the evader's best response against the strategy of pursuers is a very hard sparse-reward problem, which involves finding an escape path from the initial location to an exit node. To simplify this problem, almost all existing algorithms use the following best response approach of the evader: The evader first chooses an exit node and then randomly takes a simple (acyclic) path that guarantees reaching the chosen exit node before exceeding the maximum time horizon. This approach reduces the strategy space of the evader but cannot provide the true best response strategy for the evader. In addition, due to the above-mentioned challenge, almost all existing algorithms assume that the maximum length of the evader paths is short during training. Then, the strategy of pursuers may be exploited if the evader takes a longer path.

**The Scalability Issue of Existing Algorithms**. From the first set of experiments above, we can see that the existing algorithms require several hours to converge for solving small games, as shown in Table 1. For solving this large game with a $100 \times 100$ grid network, we cannot see reasonable results after training several days. For example, NSG-NFSP and Grasper get nothing after running four days; NSGZero and Pretrained PSRO were trained for some iterations after running four days, but their worst-case rewards are still almost 0.

These results show that existing algorithms still suffer performance and scalability issues in real-world settings, which suggest that substantial efforts are still required to develop effective and efficient algorithms for solving real-world UNSGs.

## 5 RELATED WORKS

Game theory has emerged as a valuable tool in addressing complex interactions and has been successfully applied to various security challenges (Jain et al., 2011; McCarthy et al., 2016; Sinha et al., 2018), including allocating limited resources to protect infrastructure (Jain et al., 2013) or designing patrolling strategies in adversarial settings (Vorobeychik et al., 2014). Behind these results, one important model is Stackelberg Security Games (SSGs), which is used to solve a variety of security problems (Sinha et al., 2018). In SSGs, the defender moves first and then the attacker best responds to the defender's strategy. Then, the UNSG model is a special case of SSG, which is used in the zero-sum environment on networks.

The UNSG is similar to pursuit-evasion games (Parsons, 1976), where pursuers chase evaders. The pursuit-evasion game involves strategic interactions between multiple pursuers and one or more evaders within a well-defined environment (Bilgin & Kadioglu-Urtis, 2015), presenting enduring

| Algorithm | Training | Maximum length of paths for evaluation | | | |
|---|---|---|---|---|---|
| | | $T = 6$ | $T = 6$ | $T = 12$ | $T = 12$ |
| Pretrained PSRO | 2h/1.5h | 0.01 | 0.93 | 0.01 | 0.92 |
| Grasper | 7.7h/2.5h | 0.12 | 0.97 | 0.05 | 0.95 |
| NSG-NFSP | 5.5h/3.3h | 0.03 | 0.59 | 0.03 | 0.57 |
| NSGZero | 18h/18h | 0.03 | 0.06 | 0.03 | 0.05 |
| NSGZero ($T = 3$) | 11.3h/7h | 0.01 | 0.32 | 0.0 | 0.19 |
| NSGZero ($5 \times 5$) | 1.8h/1h | 0.42 | 1 | 0.39 | 0.94 |
| CFR-MIX ($5 \times 5$) | 6.9h/6.3h | 0.03 | 0.38 | 0.01 | 0.16 |
| **Ground Truth** | hard(0.5)/easy(1) | 0.5 | 1 | 0.5 | 1 |

Table 1: The performance of existing algorithms in the worst-case setting. For the grid network, the maximum length of the evader's paths for evaluation is $T = 4$ or $T = 8$.

challenges and significant applications ranging from civilian safety (Oyler et al., 2016) to military operations (Vlahov et al., 2018). As a complex and widely-studied research problem, the pursuit-evasion game has been extensively applied across physics (Isaacs, 1965), mathematics (Pachter, 1987; Kopparty & Ravishankar, 2005), and engineering (Eklund et al., 2011). The pursuit-evasion games are often studied in the framework of differential games. Several canonical pursuit-evasion games were first formulated as differential games and extensively studied by Rufus Isaacs in his masterpiece "Differential Games" (Isaacs, 1965). Later, many studies focusing on pursuit-evasion games emerged, and different algorithms were developed. For example, Ho et al. introduced the linear–quadratic differential game (LQDG) formulation to address pursuit-evasion problems (Ho et al., 1965). In 1976, Parsons first used graphs to describe the pursuit-evasion games (Parsons, 1976). From the origins of the pursuit-evasion games until today, the game underwent several changes and now constitutes a large family of problems. Researchers have also focused on pursuit-evasion games in a discrete setting in the past several decades. The discrete-time multiple-pursuer single-evader game is solved (Bopardikar et al., 2008). Later, there are several works (Horák & Bošanský, 2016; Horák et al., 2017; Horák & Bošanský, 2017) focusing on one-sided partially observable pursuit-evasion games, in which the evader knows the pursuers' locations while the pursuers do not know the evader's location. Similarly, the patrolling security game (PSG) (Basilico et al., 2009; Vorobeychik et al., 2014), where the defender defends against an unseen intruder, and the intruder needs multiple turns to perform the attack in the environment, is typically modeled as a stochastic game with an infinite horizon. Later, PSGs were extended to cover cases where the defender receives an uncertain signal after being attacked and then goes to the point of being attacked to catch the attacker (Basilico et al., 2017a;c). More recently, a hierarchical framework has been presented for solving discrete stochastic pursuit-evasion games in large grid worlds (Guan et al., 2022). Our GraphChase can be extended to cover these settings.

Existing multiplayer benchmarks based on pursuit-evasion games, such as SIMPE (Talebi & Simaan, 2018), Multi-Agent RL Benchmark (MARBLER) (Jain et al., 2011), and Avalon (Albrecht et al., 2022), have significantly advanced the field by offering diverse scenarios and testing environments. SIMPE, for instance, focuses on interactive simulation with varied strategies for multiple pursuers and a single evader, allowing for the exploration of cooperative and non-cooperative tactics (Talebi & Simaan, 2018). However, it outputs the coordinates of the pursuer and evader in the x-y plane, with a continuous position space. And it does not take time information into account, overlooking the temporal constraints inherent in UNSGs. Similarly, MARBLER integrates physical robot dynamics with Multi-Agent Reinforcement Learning (MARL), bridging simulation with real-world robot behavior (Jain et al., 2011). Avalon further extends these concepts by providing procedurally generated worlds aimed at testing the generalization capabilities of RL algorithms (Albrecht et al., 2022). However, It is designed to simulate biological survival skills (from basic actions like eating to complex behaviors like hunting and navigation). Google Research Football (Kurach et al., 2020) and Starcraft (Samvelyan et al., 2019) are MARL environments on a plane. Despite these advances, these platforms primarily concentrate on MARL from an algorithmic development perspective, often neglecting the nuanced game-theoretical aspects that can emerge in pursuit-evasion contexts. Openspiel (Lanctot et al., 2019) is an established extensive collection of environments and algorithms for research in games. However, it mainly focuses on recreational games and does

not include pursuit-evasion games. Therefore, it results in a gap where the strategic, competitive, and cooperative elements integral to real-world applications of UNSGs need to be fully explored or optimized. Our GraphChase platform bridges the gap by building a flexible UNSG environment.

## 6 DISCUSSION: TESTBED FOR MULTIPLAYER GAMES

Computing an NE in multiplayer games is generally hard (Chen & Deng, 2005; Zhang et al., 2023b), and designing efficient algorithms for computing such an NE is still an open challenge. Our platform could be a testbed for algorithms for solving multiplayer games. In particular, our platform provides real-world scenarios for adversarial team games (von Stengel & Koller, 1997; Basilico et al., 2017b; Celli & Gatti, 2018; Farina et al., 2018; Zhang & An, 2020a;b; Zhang et al., 2021; Farina et al., 2021; Zhang et al., 2022c;a;b; Zhang & Sandholm, 2022; Carminati et al., 2022; Zhang et al., 2023a; McAleer et al., 2023; Anagnostides et al., 2023; Li et al., 2023b), where a group of players competes against an adversary or another team. Various solution concepts apply depending on the situation. When team players compete independently against the adversary, the relevant solution concepts include 1) NE (Nash, 1951; Zhang et al., 2023b), where no player gains by deviating from this equilibrium, and 2) team-maxmin equilibrium (TME) (von Stengel & Koller, 1997; Basilico et al., 2017b; Celli & Gatti, 2018; Zhang & An, 2020a;b; Zhang et al., 2022c), which is a type of NE that optimizes the team's utility across all NEs. Based on our platform, if we set that pursuers independently try to interdict the evader, we can also use our platform to compute an NE or TME in normal-form or extensive-form games. For normal-form games where team players can coordinate their strategies, the applicable solution concept is the correlated team-maxmin equilibrium (CTME) (Basilico et al., 2017b). This is essentially equivalent to an NE in zero-sum two-player games, as the team with coordinated strategies and a unified payoff function behaves like a single player. In extensive-form games, the team with coordinated strategies has two solution concepts: 1) team-maxmin equilibrium with a communication device (TMECom) (Celli & Gatti, 2018), applicable when the team can continuously communicate and coordinate strategies, making the game akin to a two-player zero-sum game with perfect recall; and 2) team-maxmin equilibrium with a coordination device (TMECor) (Celli & Gatti, 2018; Zhang et al., 2021; 2024), used when the team can only coordinate strategies before gameplay, rendering the game similar to a two-player zero-sum game with imperfect recall. The algorithms in (Zhang et al., 2019; Li et al., 2021; Xue et al., 2021; 2022; Li et al., 2023a; 2024) implemented on GraphChase compute a TMECom that is NE in team adversarial games. If we set that the team can only coordinate strategies before gameplay in the extensive-form games, we can also compute a TMECor on GraphChase.

## 7 CONCLUSION

We present GraphChase, an open-source platform for UNSGs, offering researchers a flexible multiplayer game environment to aid in developing scalable algorithms. Specifically, we first develop a unified and flexible UNSG environment and then implement several deep learning-based algorithms as benchmarks. Finally, we evaluate GraphChase and the results will provide insights into these algorithms and further refine instruction for them. We hope our GraphChase platform can facilitate the establishment of a standardized criterion for evaluating and improving algorithms for UNSGs, thereby contributing to the advancement of theoretical research and practical applications for solving general multiplayer games.

**Limitation.** Although we have implemented some state-of-the-art algorithms for solving UNSGs, these algorithms still face significant challenges on performance and scalability. As the size and complexity of the UNSG increase, computing the best response strategy for each player becomes increasingly time-consuming and computationally expensive. Existing algorithms struggle to scale up as they typically require multiple computations of the best response strategy, which can be resource-intensive. Our GraphChase platform has been designed to facilitate to address these challenges by providing a large-scale game environment. However, despite its advanced capabilities, our platform still has some limitations that we aim to address in future works. First, the abstract nature of graph-based models may not accurately capture all the dynamic and unpredictable elements of real-world environments, such as variable traffic patterns and spontaneous human behaviors. Second, GraphChase may struggle to adapt to rapid changes in urban settings, such as emergencies or unexpected social events, which can alter game dynamics and require immediate strategic adjustments.

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

# A   UNSGs in Experiments

UNSGs for the first two sets of experiments are shown in Figures 5 and 6.

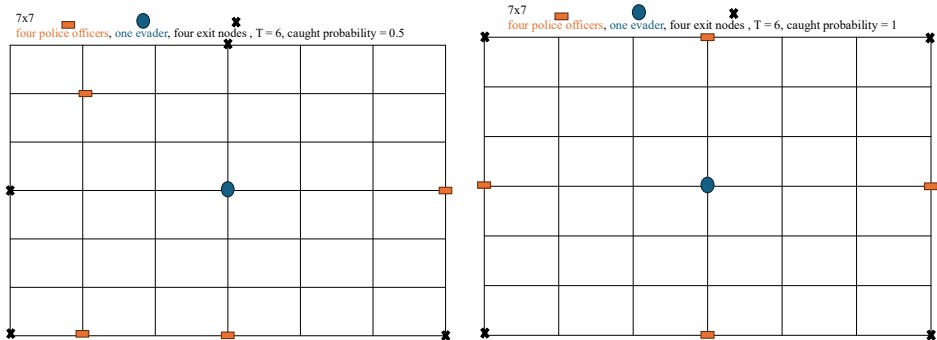

Figure 5: UNSGs of $7 \times 7$ with the caught probability of 0.5 (left) or 1 (right).

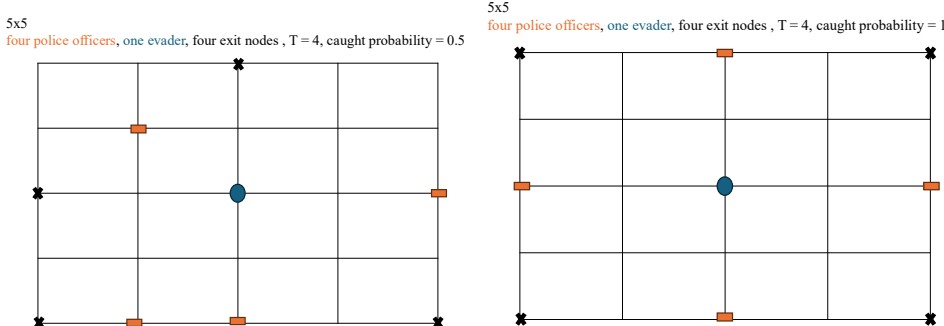

Figure 6: UNSGs of $5 \times 5$ with the caught probability of 0.5 (left) or 1 (right).

# B   Additional Experimental Results

Additional experimental results are shown in Figures 7 and 8.

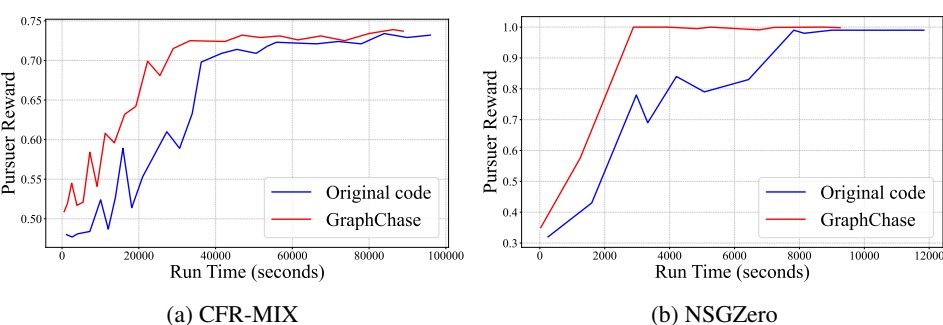

(a) CFR-MIX                    (b) NSGZero

Figure 7: The training procedure on the easy game on the $5 \times 5$ network with a caught probability of 1.

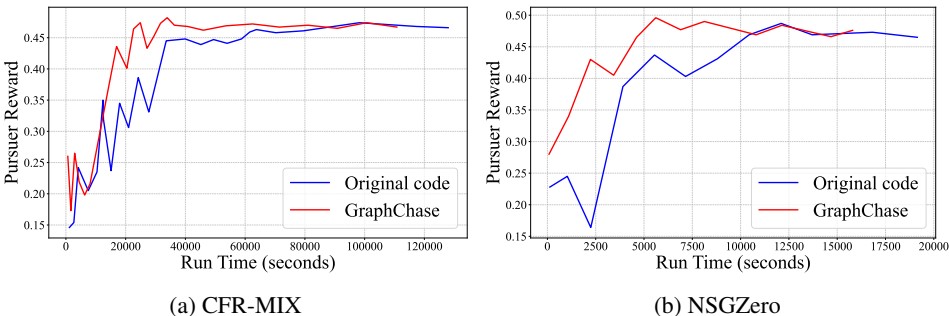

(a) CFR-MIX

(b) NSGZero

Figure 8: The training procedure on the hard game on the $5 \times 5$ network with a caught probability of 0.5.

| | Grid Graph | Custom Graph |
|---|---|---|
| underlying graph structure | column
row
side_exist_prob
diagnoal_exist_prob | adjacency matrix |
| agent number and position | max_time_horizon
pursuer_num
evader_num
exit_num
pursuer_initial_position
evader_initial_position
exit_position | |

Table 2: The parameters that users can control.

## C  USER-CONTROLLABLE PARAMETERS

The user-controllable parameters are shown in Table 2. In our platform, users can configure a range of parameters depending on the type of graph utilized: **Grid Graph** or **Custom Graph**. For the **Grid Graph**, the underlying graph structure can be controlled through parameters such as `column` and `row`, which define the grid's dimensions, as well as `side_exist_prob` and `diagonal_exist_prob`, which determine the probabilities of edges existing between adjacent nodes and diagonal nodes, respectively. For the **Custom Graph**, the underlying structure is specified via an `adjacency matrix`, allowing users to define a completely customized graph topology.

In both graph types, users can also control parameters related to the **agent number and positions**, including `max_time_horizon`, which defines the maximum simulation duration; `pursuer_num` and `evader_num`, specifying the number of pursuer and evader agents; and `exit_num`, which sets the number of exits in the graph. Additionally, initial positions for agents and exits can be customized through `pursuer_initial_position`, `evader_initial_position`, and `exit_position`, enabling users to tailor the simulation to specific scenarios.

## D  EXPERIMENTS ON OTHER SETTINGS

We conducted experiments on a $15 \times 15$ grid graph to evaluate the performance of our platform in comparison to existing environments. While CFR-MIX (Li et al., 2021), NSG-NFSP (Xue et al., 2021), and NSGZero (Xue et al., 2022) utilize the $15 \times 15$ grid graph, we found that specific settings, including the positions of pursuers, the evader, and exits, were not clearly given in their works. To ensure a fair evaluation, we adopted uniform settings for training policies across both the original code and GraphChase. There are four pursuers and ten exits for an evader. The max time horizon is 15. The same settings allow for a direct comparison of the effectiveness of our platform against the original paper.

We also extracted two real-world maps of Singapore with 372 nodes and Manhattan with 620 nodes and developed two large-scale UNSGs based on these maps. Experiments conducted on the Singapore map have been previously tested in NSG-NFSP (Xue et al., 2021), NSGZero (Xue et al., 2022), Pretrained PSRO (Li et al., 2023a), and Grasper (Li et al., 2024). Manhattan map was tested in NSG-NFSP (Xue et al., 2021), and NSGZero (Xue et al., 2022). However, specific settings for these two maps were not detailed in prior studies. For our simulations, we designated four pursuers and ten exits for the evader, with a time horizon set to 15 on the Singapore map. And there are six pursuers and ten exits for the evader, with a time horizon set to 15 on the Manhattan map. To ensure a fair comparison, we adopted the same settings for the original code and GraphChase[3]. The results are shown in the Table 3 and Table 4.

|  |  | NSG-NFSP | NSGZero | Pretrained PSRO | Grasper |
|---|---|---|---|---|---|
| $15 \times 15$ | Original paper | 0.83±0.028 | 0.87±0.021 | 0.994±0.003 | 0.995±0.002 |
|  | GraphChase | 0.85±0.021 | 0.91±0.016 | 0.996±0.002 | 0.996±0.001 |
| Singapore | Original paper | 0.92±0.027 | 0.96±0.015 | 0.996±0.001 | 0.998±0.01 |
|  | GraphChase | 0.94±0.022 | 0.97±0.014 | 0.997±0.001 | 0.998±0.01 |

Table 3: Experiments on $15 \times 15$ gird graph and real-world map from Singapore. Approximate worst-case defender rewards, averaged over 1000 test episodes. The "±" indicates 95% confidence intervals over the 1000 plays.

|  | NSG-NFSP | NSGZero |
|---|---|---|
| **GraphChase** | $0.8689 \pm 0.1377$ | $0.8865 \pm 0.0859$ |
| **Original Code** | $0.8556 \pm 0.1151$ | $0.8738 \pm 0.1377$ |

Table 4: Experiments on real-world map from Manhattan. Approximate worst-case defender rewards, averaged over 1000 test episodes. The "±" indicates 95% confidence intervals over the 1000 plays.

# E  FASTER WALL-CLOCK CONVERGENCE

Our platform incorporates several technical enhancements that contribute to its faster performance. First, we have adopted the Gymnasium for game simulation, replacing the custom class implementations found in the original papers. This change results in faster simulation processes and eliminates redundant data copying operations, leading to improved efficiency.

Additionally, we have implemented various code optimizations to enhance the platform's performance. These include improved data type conversions, such as using numpy-to-tensor conversions instead of list-to-tensor operations, which reduces processing time. We have also focused on enhancing memory management throughout the platform, resulting in more efficient resource utilization.

From the perspective of wall-clock time, this indeed accelerates the convergence speed. However, it's crucial to note that in terms of the number of training iterations required for convergence, there is no significant improvement. For instance, if the original code necessitates sampling $10^4$ episodes to initiate convergence, our platform's reproduced algorithms similarly require approximately the same number of training iterations. This consistency in training iterations is attributable to the fact that we have not altered the underlying algorithms themselves.

Unlike the original implementation, our platform is designed with modular components, making it unsuitable to directly compare the performance of individual components against the original code. However, to emphasize the efficiency of our platform in simulation processes, we conducted experiments to evaluate the time required for a single episode of simulation and the subsequent data-saving process for each algorithm. The performance comparison between GraphChase and the

---

[3]Due to the extended training time required for the CFR-MIX algorithm, we did not conduct tests for CFR-MIX.

original implementation, highlighting the significant speed improvements achieved by our platform, is presented in Table 5.

|  | NSG-NFSP | NSGZero | Pretrained PSRO | Grasper |
|---|---|---|---|---|
| **GraphChase** | $0.0089 \pm 0.005$ | $0.378 \pm 0.12$ | $0.0065 \pm 0.002$ | $0.0097 \pm 0.002$ |
| **Original Code** | $0.0187 \pm 0.005$ | $0.523 \pm 0.15$ | $0.0153 \pm 0.004$ | $0.0178 \pm 0.002$ |

Table 5: Performance comparison between Original Code and GraphChase in terms of simulation and data-saving time (in seconds). Each value represents the mean execution time for a single episode, with the corresponding standard deviation shown after the symbol $\pm$.

## F USAGE INSTRUCTIONS FOR GRAPHCHASE

The following steps outline the process for setting up and utilizing the GraphChase platform:

### F.1 CLONING THE REPOSITORY

To begin, clone the GraphChase repository from GitHub and navigate to the project directory:

```
git clone https://github.com/GraphChase/GraphChasePlatform.git
cd GraphChasePlatform
```

### F.2 INSTALLING DEPENDENCIES

Install the necessary dependencies including pytorch, DGL and other required dependencies

### F.3 RUNNING AN ALGORITHM

To run a specific algorithm, such as NSGZero, perform the following steps:

1. **Customize the Graph:** Modify the graph file located at `graph/graph_files/custom_graph.py` to configure the graph structure, as well as the positions of pursuer, evader, and exits.

2. **Adjust Algorithm Parameters:** Open the configuration file in the `configs` directory, such as `nsgzero_configs.py`, and set the desired parameters.

3. **Run the Algorithm:** Execute the script to run the NSGZero algorithm:

   ```
   python scripts/run_nsgzero_solver.py
   ```

The procedure for executing other algorithms follows a similar structure, requiring adjustments to their respective configuration files and script execution.

## G REPRODUCIBILITY

The structure of the network and values of all parameters follow the original papers of our implemented algorithms. To ensure the fairness of the comparative experiments, all our experiments were conducted on the server with 48-core 3.00GHz Intel(R) Xeon(R) Gold 6248R CPU and 8 NVIDIA A30 GPUs.

We release our platform on: https://github.com/GraphChase/GraphChasePlatform.git

