# OpenReview forum: "Solving Urban Network Security Games: Learning Platform, Benchmark, and Challenge for AI Research"
_ICLR.cc/2025/Conference — Submitted to ICLR 2025_

### Official Review · Reviewer_HG6y · 2024-11-02

**Soundness:** 3
**Presentation:** 3
**Contribution:** 2
**Rating:** 5
**Confidence:** 3

**Summary:**

The paper proposes GraphChase, a Gymnasium-based platform that is aimed to solve urban network security games (e.g., policy trying to catch a criminal/terrorist in a densely populated urban environment where avoidance of collateral damages and civilian casualties is crucial). The platform has a modular structure allowing the user to define the environment (by entering a - potentially complex - graph), the strategies of the players and the learning algorithms among other parameters. The paper runs experiments in 7x7 grids with 4 police officers and 1 criminal and demonstrates that solution through GraphChase achieves generally faster solution times than naive implementation of the learning algorithms.

**Strengths:**

- *Quality and Clarity*: the paper is well-structured and cover many details of the proposed GraphChase platform. There is a discussion of related work, potential extensions to multi-agent settings and limitations of the current work which mainly stem from the (convergence) limitations (in complex environments) of state-art MARL algorithms. The exposition is generally clear, and I could follow the paper.

- *Originality and Significance*: the problem of urban network security games is important - entails optimisation of police strategies to arrest criminals which becomes particularly challenging in urban environments were casualties of co-existing civilians need to be avoided at any cost - and providing a platform to make their solution easier is an ambitious and well-received goal. There is a literature that studies this topic, and although I am not expert in this literature, the paper does a good job to present it for the general reader and motivate the current study.

-*Reproducibility*: The authors provide a link to a github repository with their code which I appreciated.

**Weaknesses:**

- *Contribution and Novelty*: the main weakness in my opinion is that the paper does not provide convincing enough evidence that GraphChase is at the moment a significant step in solving UNSGs. The algorithmic results provide some evidence that it converges faster, but this is not surprising since the platform and the learning algorithms are integrated. I am not sure that I understand correctly the reasons of the claimed speed-up (so, please see my question below). Also, some of the simulations seem to terminate early or do not demonstrate significant improvements over the naive algorithm.

By weighting the strengths and weaknesses, my evaluation is that while I don't see methodological errors or bad exposition, the contribution of the current paper is simply not enough to merit publication at its current stage at ICLR. The paper should provide more substantial experiments in more complex environments and more systematic comparisons to the existing literature. While I understand that such environments can grow very complex very quickly demanding unrealistic computational power to be solved on researchers' computers, I still believe that the current contribution is not sufficient.

**Questions:**

- Can the authors clearly explain how is the speed-up achieved in the simulations that they show? In other words, what are the exact reasons/mechanisms that make the solution through GraphChase faster than the "naive" solution. Also how does the "original code" (or naive) implementation solve the games? Can the authors provide a detailed breakdown of computational time for different components of their algorithm compared to the baseline approaches?
- Why do some simulations seem to terminate early (especially the red lines) and some others to terminate before convergence (e.g., first panels of Fig 7 and 8). Can you please explain the termination criteria in detail, and provide convergence plots that show the full trajectory of the algorithms until a well-defined convergence point is reached?
- Can the authors provide specific additional experiments in already existing environments from the literature (with references) and  comparisons that would provide more convincing evidence of GraphChase's significance?

---

> ### Author Response · Authors · 2024-11-23
>
> We appreciate the reviewer's thoughtful critique and the opportunity to clarify our platform's contributions.
>
> ## **To Weaknesses:**
>
> The primary goal of GraphChase is not to propose a novel algorithm or demonstrate superior performance than existing work, but rather to provide a unified, open-source environment for training and testing for UNSGs. Our platform enables researchers to efficiently utilize existing algorithms, including NSGZero, NSG-NFSP, PretrainPSRO, and Grasper, for performance evaluation. This addresses a critical gap in solving UNSGs, which is why we submitted to the Datasets and Benchmarks track.
>
> Since GraphChase aims to provide a unified simulation environment for existing algorithms, we focused on reproduction rather than algorithmic improvements. That's the reason GraphChase does not have significant improvements over the original algorithm. We optimized code structure and implementation, resulting in faster wall-clock convergence compared to the original implementations. This aligns with our objective of achieving comparable performance to the original implementations.
>
> Concerning the systematic comparisons to the existing literature, we deliberately matched the experimental settings of the original papers to validate GraphChase's ability to reproduce their results. As our primary goal is to demonstrate equivalent performance to original implementations rather than surpassing them, we suppose our current experiments sufficiently validate this objective.
>
> Our experiments also show that current algorithms still suffer performance and scalability issues in real-world settings. It suggests that substantial efforts are still required to develop effective and efficient algorithms for solving real-world UNSGs.
>
> GraphChase supports a wide range of equilibrium concepts, including, but not limited to Nash Equilibrium. The platform allows researchers to explore and analyze diverse equilibrium types by integrating their custom solution algorithms. This flexibility distinguishes GraphChase from existing platforms, which typically lack support for such extensibility.
>
> We look forward to the research community utilizing GraphChase as a collaborative tool for advancing UNSG research. Furthermore, we will provide more details about why it is faster and when to terminate in response to your subsequent questions.
>
> ## Questions:
>
>
> - **To Question 1:** We can detail the specific technical factors contributing to the improved performance:
>
>    Our platform incorporates several technical enhancements that contribute to its faster performance in terms of the wall-clock time. First, GraphChase is developed based on Gymnasium, replacing the custom class implementations found in the original papers. This change results in faster simulation processes and eliminates redundant data copying operations, leading to improved efficiency.
>
>    Additionally, we have implemented various code optimizations to enhance the platform's performance. These include improved data type conversions, such as using numpy-to-tensor conversions instead of list-to-tensor operations, which reduces processing time. We have also focused on enhancing memory management throughout the platform, resulting in more efficient resource utilization.
>
>    Currently, different algorithms (such as NSGZero, PretrainPSRO, and Grasper) each implement their simulation environments differently. This requires researchers to rewrite substantial code to adapt their algorithms for comparison with existing methods. When using the original code to test games, we rewrote game-related code to fit the input data requirements of each algorithm. However, we did not make any modifications to the algorithmic implementations from the original papers.
>
>    The algorithms implemented in our platform are direct reproductions of the original code provided by the respective papers, with no changes or improvements made to the algorithms themselves. The key difference is that in GraphChase, researchers only need to input graph adjacency list as parameters into the game generation function to enable the application of different algorithms. Therefore, the primary contribution of GraphChase lies in providing a standardized and unified testing environment, addressing the lack of platforms for UNSG research.
>
>    To address the reviewer's concern about performance improvements, we added a detailed discussion in Appendix E comparing GraphChase's simulation speed with original implementations. The results show that GraphChase truly accelerates a single episode of simulation and the data-saving process for each algorithm.
>
>    Hope our reply will clarify your concerns.

---

> ### Author Response · Authors · 2024-11-23
>
> ## Questions:
>
> - **To Question 2:** Thank you for this important observation regarding the termination patterns in Figures 7 and 8. We acknowledge that this requires clarification.
>
>    For all experiments, including both GraphChase and baseline implementations, we set a large maximum iteration count at the start of training. The earlier termination of the red lines in the figures is a direct result of GraphChase's improved computational efficiency as we explain above, allowing it to complete the whole iterations more rapidly than the baseline implementations.
>
>    Regarding the specific cases presented in the first panels of Figures 7 and 8, we acknowledge that the data shown in our original submission only covered the results up to the point of convergence. Data after this point, which would have depicted a more stable convergence curve, was not included. We recognize that it may not have provided a complete view of the algorithms’ convergence trajectories. To address this, we have updated our results to include a more comprehensive depiction of the convergence process. We apologize for any confusion this omission may have caused and appreciate the opportunity to present more complete and accurate experimental results in Figures 7 and 8.
>
> - **To Question 3:** We added experiments based on a $15\times 15$ grid graph and the Singapore map. The $15\times 15$ grid graph has been previously tested in CFR-MIX (Li et al., 2021), NSG-NFSP (Xue et al., 2021) and NSGZero (Xue et al., 2022). The Singapore map has been tested in NSG-NFSP (Xue et al., 2021), NSGZero (Xue et al., 2022), Pretrained PSRO (Li et al., 2023a) and Grasper (Li et al., 2024). These additional experiments can show the significance of GraphChase. The experimental results are detailed in Table 3 of the appendix.

---

> > ### Comment · Reviewer_HG6y · 2024-11-25
> >
> > I thank the authors for their detailed response. I can know better appreciate their contribution. The new experiments in the $15\times15$ Singapore map which recover the performance of existing algorithms (Table 3) but at half the running time (Table 4) -- if I understand correctly -- strengthen the paper. Similarly, the instructions on how to use the platform in Appendix F are also welcome. I raise my score to reflect these changes. Although Datasets and Benchmarks is not my area of research (I focus on theory), and after reading the other, more expert, reviews, I still believe that more systematic experiments in more complex environments (the above being correct steps in that direction) and a more careful presentation perspective to highlight the key contributions of the platform (the above being again correct steps in that direction) are required.

---

> > > ### Author Response · Authors · 2024-11-28
> > >
> > > Thank you for your constructive feedback!
> > >
> > > We have supplemented additional experiments based on the Manhattan map. Now we have additional experiments on a variety of scenarios used in the UNSG domain (Xue et al. 2021; 2022; Li et al. 2023a; 2024) in Section 4.2 (details are in Appendix D), which includes larger $15\times 15$ grid structures and real-world maps based on Singapore and Manhattan. The $15\times 15$ grid network represents the randomly generated network, and two real-world networks represent different topological structures in real-world cities.  To the best of our knowledge, the Singapore and Manhattan maps represent the most realistic and complex graph structures currently employed in UNSG algorithm research. We believe that our extensive testing on these maps effectively demonstrates GraphChase's capability to address real-world problems.
> > >
> > > We have revised the manuscript to more clearly highlight the key contributions of our platform in Sections 1 and 4.2. Specifically, we highlight that algorithms based on GraphChase can recover the performance of the algorithms based on the original codes with significantly less time. The modifications have been marked in red for your convenience, and you can review them in the latest version of the paper.

---

> > > ### Author Response · Authors · 2024-12-01
> > >
> > > Thank you for your previous response. We would appreciate your feedback on whether our additional response has adequately addressed your concerns. If there are any remaining questions or points that need further clarification, we would be happy to provide additional information.

---

> > > > ### Comment · Reviewer_HG6y · 2024-12-03
> > > >
> > > > I thank the authors for their response and I maintain my current assessment for the paper.

---

### Official Review · Reviewer_eAes · 2024-11-03

**Soundness:** 3
**Presentation:** 2
**Contribution:** 2
**Rating:** 5
**Confidence:** 4

**Summary:**

The paper introduces GraphChase, a scalable, open-source platform tailored for Urban Network Security Games (UNSGs), where police agents coordinate to capture criminals in complex urban settings. UNSGs present computational challenges due to their large game spaces and the need for cooperation and competition among multiple players under imperfect information. Existing algorithms like CFR-MIX and PSRO have struggled with scalability and performance in these settings. GraphChase aims to provide a flexible environment for developing and benchmarking algorithms across diverse UNSG scenarios, supporting features including different numbers of players, types of underlying graphs, information levels, and the presence or absence of communication among police officers. Experimental results indicate that GraphChase improves algorithm efficiency compared to previous implementations but reveals ongoing challenges in scaling algorithms for larger urban networks​.

**Strengths:**

1. The paper addresses a challenging and practical problem—Urban Network Security Games (UNSGs)—that has direct applications in urban safety and law enforcement.
2. The paper aims to design an open-source, flexible, and modular platform that facilitates simulation, testing, and benchmarking of various algorithms in scalable settings. This makes it valuable for researchers aiming to develop efficient algorithms and compare them in a fair way.
3. The paper implements multiple algorithms in its experiments, effectively identifying performance and scalability issues.

**Weaknesses:**

Considering that this is a submission for the datasets and benchmarks track, I have several concerns about the weakness of this paper.

1. Given that GraphChase is claimed as a comprehensive platform enabling various configurations, the experiments seem limited. Additional experiments and result analysis on different underlying graphs and varying information levels would better demonstrate the platform’s versatility and advantages.
2. It seems that the platform only considers algorithms for finding the optimal strategy for the pursuer, and directly computes the criminals' best responses as evaluation. However, the paper does not discuss the computational efficiencies of evaluation.
3. The paper elaborates related work on the pursuit-evation games and mentions their relationships to UNSGs. However, first, I found the literature comparison in this part is unclear. Second, the paper does not mention the relationships in detail nor does it illustrate using experiments. For example, does UNSG include other games beyond pursuit-evasion games?
4. The paper does not discuss the efficiency of using and interacting with the platform.
5. The experimental results are not presented in a clear way, as detailed in the "Questions" section below.

**Questions:**

1. In Line 347, how is the probability of being caught calculated? Specifically, I do not understand why the left side of Figure 5 has a probability of 0.5.
2. In Section 4.2, the authors aimed to show that their reproduced code performs better than the original. Could the authors further clarify this comparison? Is the improvement due to enhancements in code details, or does GraphChase offer systemic advantages that enable faster algorithm convergence?
3. In Table 1, the paper notes that even in relatively small settings, existing methods require extensive training times. Is this due to the complexity of environment interactions? Does this imply a lack of efficiency in the current platform, or is it simply a limitation of the algorithms themselves?
4. Around Line 410, the authors mention that current methods often make simplified assumptions about the criminal's strategy. Would it be possible to observe phenomena or convergence patterns if both the criminal and the police officer were modeled as learning algorithms?

---

> ### Author Response · Authors · 2024-11-23
>
> Thank you for your thoughtful review! Please find point-by-point replies below:
>
> ## Weaknesses:
> **To Weakness 1:** We would like to clarify two key aspects of GraphChase:
>
>    Regarding information levels, our platform allows researchers to configure game environments according to their algorithmic requirements. This flexibility is demonstrated in our current implementations: CFR-MIX makes decisions without considering evader history, while PretrainPSRO incorporates historical evader information. This exemplifies GraphChase's capability to support varying information levels. That is the information level requirements are determined by the algorithms themselves. Currently, there are only a few learning algorithms for UNSGs and the results of them are shown in our work.
>
>    Concerning graph structures, GraphChase supports the removal of edges between any connected nodes to simulate different real-world scenarios. This feature has been tested with the Grasper algorithm, as it requires the generation of diverse graph structures. Furthermore, GraphChase allows researchers to customize various graph structures by providing an adjacency list as a parameter to our game generation function.
>
>    We designed GraphChase to offer maximum flexibility through user-defined graph structures and customizable information levels. We hope that our platform serves as a versatile tool for researchers, enabling them to efficiently implement and evaluate various algorithms.
>
> **To Weakness 2:** We would like to clarify several aspects of our platform's implementation and evaluation methodology:
>
>    Regarding finding the evader's optimal strategy, our current implementations follow the simplified evader modeling approach used in existing UNSG research. This simplification means that it only considers finding the optimal strategy for the pursuer, and directly computes the criminals' best responses as evaluation.
>
>    The computational complexity of UNSGs presents significant challenges, as the strategy space of players cannot be enumerated even under simplified conditions where the time dynamics are ignored (Jain et al., 2011). That is, even if the evader's strategy is simplified, the problem of solving UNSGs is still NP-hard (Jain et al., 2011), which is discussed in Section 2.3 of the paper.
>
>    We also note the limitations of solely computing the pursuer's optimal strategy during training, as evidenced by the experimental results in Table 1. The pursuer's performance notably deteriorates when confronting evader strategies not encountered during training. This observation was one of the key motivations behind developing GraphChase. So our platform enables researchers to model evaders with any strategy. Specifically, GraphChase allows researchers to customize evader decision-making mechanisms such as implementing them as learning algorithms to find optimal strategies.
>
>    Due to the computational complexity in evaluation, we sample 1000 episodes to calculate the pursuers' worst case utility. At the beginning of each episode, the evader selects an available path based on the best response i.e., enumerate all paths to select the best path. During each time step, the evader moves according to this predetermined path. The pursuer's performance is evaluated based on the average success rate across these 1000 episodes.
>
>    As we discussed in Section 2.3, scalability is the key challenge for solving UNSGs. Our GraphChase platform has been designed to facilitate and address these challenges by providing a large-scale game environment.

---

> ### Author Response · Authors · 2024-11-23
>
> ## Weaknesses:
>
> **To Weakness 3:** We would like to clarify the scope and positioning of UNSGs within the pursuit-evasion games domain.
>
>    While pursuit-evasion games encompass various scenarios, UNSGs represent a specific subset with distinct characteristics. UNSGs are specifically focused on scenarios with discrete observation and action spaces, chasing within a finite time horizon. In contrast, existing pursuit-evasion game benchmarks like Avalon, mentioned in our related work, are designed to simulate biological survival skills (from basic actions like eating to complex behaviors like hunting and navigation). In the revised version, we added more details about existing platforms SIMPE and Avalon, and highlighted the difference between them and our platform. Its problem settings do not belong to UNSGs.
>
>    Our related work discussion (L437-460) examines various pursuit-evasion scenarios that can be modeled as UNSGs. These works, while valuable, do not provide a platform for UNSGs. So we do not conduct experimental comparisons with GraphChase.
>
>    Regarding existing benchmarks (discussed in L462-478), we provide a brief overview of these benchmarks and explain why they are not suitable for UNSGs. For instance, MARBLER primarily simulates real-world robot behavior, while SIMPE operates in continuous state space and imposes restrictions on the behavior patterns of both pursuers and evaders. These limitations make existing benchmarks unsuitable for studying UNSGs.
>
>    So the motivation behind developing GraphChase is precisely to address this gap by providing a platform and benchmark tailored for research on UNSGs. Thank you for your suggestion. We added more comparisons in the related work to clarify the differences between GraphChase and the current platforms for pursuit-evasion games, which we believe will aid in the readers' understanding.
>
> **To Weakness 4:** We appreciate the reviewer's comment about platform usability and interaction documentation.
>
>    We deliberately chose Gymnasium as the foundational framework for GraphChase because it is widely recognized and utilized in the DRL community. This design choice significantly reduces the learning curve for researchers who are already familiar with Gymnasium's interfaces and conventions.
>
>    Regarding platform usage and interaction details, we have provided comprehensive documentation and examples in our GitHub repository. We added a brief introduction about how to use our platform to Appendix F.
>
>    Thank you for bringing this to our attention. These additions will make the paper more complete and useful for potential users of our platform.

---

> ### Author Response · Authors · 2024-11-23
>
> ## Questions:
>
> 1. **To Question 1:** Let us explain how we arrived at the probability of 0.5 under the Nash equilibrium:
>
>    On the left side of Figure 5, the evader has four potential exit nodes. The calculation considers several key factors:
>
>    Exit Node Accessibility:
>    - Two exit nodes (bottom-left and bottom-right) have the shortest paths longer than 3 steps for the evader
>    - These same exits can be reached by pursuers in exactly 3 steps
>    - Due to rational decision-making, the evader will avoid these exits as capture would be certain
>
>    Viable Exit Options:
>    - This leaves two viable exit nodes for the evader
>    - Only the pursuer in the top-left corner can reach either of these exits within 3 steps
>    - Both exits have equal utility for both the evader and the pursuer
>
>    Probability Calculation:
>    - The evader rationally chooses between the two viable exits with equal probability (1/2 for each)
>    - For each exit, the probability of being caught depends on the pursuer choosing the same exit (1/2)
>    - The total probability is calculated as: 2 exits × (1/2 probability of choosing each exit) × (1/2 probability of pursuer choosing the same exit) = 0.5
>
>    Hope our reply will clarify your concerns.
>
> 2. **To Question 2:** Our platform incorporates several technical enhancements that contribute to its faster performance in terms of the wall-clock time. First, we have adopted the Gymnasium for game simulation, replacing the custom class implementations found in the original papers. This change results in faster simulation processes and eliminates redundant data copying operations, leading to improved efficiency.
>
>    Additionally, we have implemented various code optimizations to enhance the platform's performance. These include improved data type conversions, such as using numpy-to-tensor conversions instead of list-to-tensor operations, which reduces processing time. We have also focused on enhancing memory management throughout the platform, resulting in more efficient resource utilization.
>
>    From the perspective of wall-clock time, this indeed accelerates the convergence speed. However, it's crucial to note that in terms of the number of training iterations required for convergence, there is no significant improvement. For instance, if the original code necessitates sampling $10^4$ episodes to initiate convergence, our platform's reproduced algorithms similarly require approximately the same number of training iterations. This consistency in training iterations is attributable to the fact that we have not altered the underlying algorithms themselves.
>
>    This distinction highlights an important nuance: while GraphChase offers improved computational efficiency in the environment, it does not change the sample efficiency or learning process of the implemented algorithms. Our platform's primary contribution lies in providing a more efficient simulation environment, rather than enhancing the algorithmic performance itself.

---

> ### Author Response · Authors · 2024-11-23
>
> ## Questions:
>
> 3. **To Question 3:** The extensive training times are primarily due to the limitations of the current algorithms rather than platform inefficiencies. Let me elaborate on some factors contributing to these computational demands:
>
>    - Grasper and Pretrained PSRO: These algorithms require extensive training to generate the best responses in each iteration. Our platform's current implementations require $10^5$ episode simulations(same as the original paper) to gather sufficient data for RL training. This sampling requirement is inherent to the PSRO methodology.
>
>    - NSGZero:
>    Similar to AlphaGo's approach, it employs Monte Carlo Tree Search. Requires extensive tree exploration to generate effective policies. The computational intensity is intrinsic to the search-based nature of the algorithm.
>
>    Other implemented algorithms similarly require substantial data sampling for effective training.
>
>    We also note that the significant training time requirements, even for relatively small games (e.g., $5\times 5$ grid), represent a broader challenge in the field of UNSGs. Actually, it is one of the reasons that we develop the GraphChase platform.
>
>    In addition, our new Appendix E shows that our platform GraphChase runs faster than existing implemented environments.
>
>    We aim to provide researchers with a standardized environment for algorithm development and testing. We hope that our platform can enable researchers to focus on algorithmic design that could potentially reduce these training times and improve computational efficiency, rather than spending time on environment implementation details.
>
> 4. **To Question 4:** The simplified assumptions about criminal strategies primarily serve to facilitate convergence analysis due to the hardness of computing the evader's best response. Modeling both criminals and pursuers as learning agents presents a practical challenge: As demonstrated in NSG-NFSP (Xue et al., 2021), when criminals are modeled as learning agents, they tend to spend considerable time exploring invalid paths (those not leading to targets) before discovering optimal strategies. This results in:
>    - Training inefficiency due to pursuers learning against suboptimal opponents
>    - System instability during the learning process
>
>    These findings have influenced subsequent works, including NSGZero, Pretrained PSRO, and Grasper, to adopt similar modeling approaches for criminal strategies. That's the reason that our current testing algorithms follow the same settings.
>
>    However, our platform fully supports the implementation of criminals as learning agents, a feature that is not supported in the original papers. Researchers can integrate their own training algorithm and criminal agent designs into our simulator. We truly hope that our GraphChase is a useful tool for advancing UNSG research by enabling the development and testing of more sophisticated criminal strategy models.

---

> > ### Comment · Reviewer_eAes · 2024-12-01
> >
> > Thank you for your detailed response.
> >
> > With respect to strategy or algorithm evaluation, does your platform support the implementation of various evaluation methods for pursuers' strategies, as well as the computation and assessment of the overall equilibrium or strategy profile? If so, could these processes be further illustrated and explained in the paper or accompanying documentation?

---

> ### Author Response · Authors · 2024-12-01
>
> Thank you for your review. We would appreciate your feedback on whether our response has adequately addressed your concerns. If there are any remaining questions or points that need further clarification, we would be happy to provide additional information.

---

> ### Author Response · Authors · 2024-12-03
>
> Thank you for your insightful question regarding strategy evaluation methods. We appreciate the opportunity to elaborate on our approach to assessing pursuer strategies and equilibrium computation.
>
> Our platform supports the pursuer's **strategy evaluation** through four methods:
>
> 1. Pseudo Worst-Case Utility
>
> This method evaluates the performance of the pursuer's strategy by first selecting an exit according to the best response and then choosing a path within that exit. Due to the randomness in path selection, it is referred to as pseudo worst-case utility.
>
> 2. Worst Case Utility
>
> This method evaluates the performance of the pursuer's strategy by enumerating every feasible path and using the path with the lowest reward as the worst-case utility. When the number of paths is vast and difficult to enumerate, it will use the first method to approximate worst-case utility.
>
> 3. Strategy Robustness Testing
>
> We assess strategy robustness by varying the maximum time horizon, which enables a comprehensive examination of the pursuer strategy's performance against diverse evader behaviors. As demonstrated in our experimental setup, this approach provides insights into the strategy's adaptability across different scenario complexities.
>
> 4. Exploitability Testing
>
> This method systematically evaluates the pursuer strategy's vulnerability by training an adversarial evader strategy using reinforcement learning while maintaining a fixed pursuer strategy.
>
> Regarding **equilibrium assessment**, our platform supports calculating the NashConv metric to measure convergence. This metric is calculated as:
>
> ```math
> NashConv = pursuer_br_value  +  evader_br_value
> ```
> , where `pursuer_br_value` and `evader_br_value` denote the values of their respective best response strategies. Since we provided the aforementioned methods for strategy evaluation, there are several ways to assess NashConv. Users can utilize (Pseudo) Worst-Case Utility or various training algorithms to calculate the value of best response strategies. The detailed usage methods can be found on [GraphChase repository](https://github.com/GraphChase/GraphChasePlatform.git).
>
> The evaluation method outlined in Section 3.2 of our paper is consistent with prior works on UNSG, including CFR-MIX (Li et al., 2021), NSG-NFSP (Xue et al., 2021), NSGZero (Xue et al., 2022), Pretrained PSRO (Li et al., 2023a), and Grasper (Li et al., 2024). To ensure a fair and direct comparison with these previous studies, we did not include alternative evaluation methods in the paper.
>
> However, our platform fully supports additional evaluation methods as mentioned above. These methods have been implemented at the code level, so we have provided detailed example instructions in the usage guide available on [GraphChase Platform](https://github.com/GraphChase/GraphChasePlatform.git). The guide provides detailed instructions on how users can leverage these various assessment techniques to evaluate strategies and analyze equilibrium characteristics.
>
> We hope this detailed explanation addresses your concerns and provides clarity on our approach to strategy and algorithm evaluation.

---

### Official Review · Reviewer_FH55 · 2024-11-03

**Soundness:** 3
**Presentation:** 2
**Contribution:** 2
**Rating:** 3
**Confidence:** 4

**Summary:**

This paper considers Urban Network Security Games (UNSGs), a game-theoretic model of a scenario in which pursuers seek to catch some evaders. The game takes place on a graph. Agents are positioned at certain nodes and can move to nodes in the neighbourhood by taking an action, and evaders can exit at a pre-specified set of nodes. The work implements a set of environments corresponding to variations in this family of games, as well as a set of algorithms that can be applied to them. Some benchmarks are shown that demonstrate that the proposed implementations are faster than those in the original papers.

**Strengths:**

S1. The basic idea behind the work (proposing a framework for developing learning algorithms for a class of multiplayer network games) is sensible and worthwhile.

S2. The work is technically sound.

**Weaknesses:**

W1. The presented benchmarks are very thin: the only aspect they demonstrate is that the authors' implementations are faster than the original codebases. In my opinion, this is a missed opportunity to showcase the type of experiments that are enabled by the existence of this framework. How do the different algorithms perform on the different game variants (e.g. pursuers able communicate versus not; the different observability of the locations as in cases i-iv described in Section 2.2; different types of graph topologies as you only consider grids). Without this type of analysis, the contribution of the paper is lacking. Including it would validate the argument that your framework would be helpful in carrying out algorithmic research.

W2. The paper reads more like an advertisement for the framework of the authors rather than objective scientific writing. The framework is self described as "advanced", "pivotal", etc. We also get sentences that seem to suggest the design is somehow highly innovative such as "[...] providing a seamless flow of information and actions across the system. This modular approach not only enhances the adaptability of the platform to different research demands and scenarios but also supports the integration of various algorithmic strategies". In my opinion, the design contains just about every component you would expect for a multiplayer game environment and is standard.

**Questions:**

C1. Could you define more precisely in Section 2.1 what it means to for the pursuer to "catch" the evader, as it seems this is currently missing? Presumably, the evader is caught if both the pursuer and evader are located at the same node, and the node is not an exit node?

C2. L131: $E_{exit}$ or preferably $E_\text{exit}$? $\mathcal{N}(v)$ denotes either the *set of neighbours* or the *neighborhood*; L173: $\mathbb{R}$ instead of $R$?

C3. In Section 2.2 you say that NE is adopted as solution concept, but Section 6 mentions that a TMECom is computed in your framework. Section 2.2 should be updated with clarifications.

C4. For the "Game Module" component (3.1), you should specify what formats for graph data are supported e.g. graphml, xml, adjacency list, etc.

C5. Regarding the presented benchmarks, could you specify *why* your implementations perform faster? What is the core insight or optimization you did that enables this? This is not discussed.

C6. This is more of a preference, but I would suggest sticking to the "pursuer" and "evader" terminology throughout the paper. The work keeps intermittently referring to police officers and criminals, but this is a highly abstracted model, and while it is inspired by a real-world scenario, it is very far from capturing real-world complexity. Applying this terminology can also help to more clearly signal that the paper does not raise any ethical or fairness concerns (I don't think it does).

---

> ### Author Response · Authors · 2024-11-23
>
> Thank you for your thoughtful review! Please find point-by-point replies below:
>
> ## Weaknesses:
> **To W1:** We would like to clarify that our work primarily aims to provide a standardized platform for UNSG research and algorithm development, that's why we submitted to the Datasets and Benchmarks track.
>
>    Current UNSG-focused works (NSGZero, NSG-NFSP, Pretrained PSRO, Grasper, and CFR-MIX) implement environments differently, requiring researchers to rewrite substantial code to adapt to each algorithm's input requirements when conducting comparative studies. GraphChase addresses this challenge by integrating multiple into our platform. Researchers need only input the graph structure once to evaluate multiple algorithms' performance. This not only enhances research efficiency but also ensures fair algorithm comparisons by resolving implementation inconsistencies across different works.
>
>    One of the key features of GraphChase is its flexibility, allowing users to customize various game variants. We believe that the definition of game variants should be determined by the algorithm designers rather than by us, in order to support different algorithms training on the same graph structure. Our platform provides all state information at each simulation step, including all pursuer and evader observations. Researchers can freely define how their algorithms utilize this information. For instance, while CFR-MIX doesn't consider evader history, PretrainPSRO incorporates historical evader information in pursuer action selection. Despite these methodological differences, our platform requires only a single input of the game's topological structure to enable training and testing across different algorithms and game variants on the same game structure.
>
>    Furthermore, regarding graph topologies, our platform supports arbitrary graph structures for UNSG problems. Researchers can define any graph topology (grid-based or otherwise) by simply providing the adjacency list to our platform's game generation function. The platform architecture is intentionally designed to accommodate various equilibrium concepts as shown in the discussion. Researchers can leverage our platform to study different types of equilibria by implementing their own algorithms.
>
>    Our experiments show that current algorithms still suffer performance and scalability issues in real-world settings. It suggests that substantial efforts are still required to develop effective and efficient algorithms for solving real-world UNSGs.
>
>    We envision GraphChase as a unified and flexible platform that will advance research progress in UNSG problems.
>
> **To W2:** We have revised the paper to adopt a more neutral and precise tone, removing subjective descriptors like "advanced" and "pivotal." Thank you for helping us maintain appropriate scientific writing standards.
>
> ## Questions:
> **To C1:** Thank you for your question regarding the definition of 'caught'. We acknowledge that we overlooked this definition in the original paper. To clarify: The evader is considered caught if the evader and any of the pursuers occupy the same point at any time within the maximum time horizon. We have now added this precise definition to Section 2.1.
>
> **To C2:** We appreciate your attention to detail.
>
>    1. Update $E_{xit}$ to $E_{exit}$.
>    2. N(v) specifically denotes the set of neighbours of node v. We have clarified this definition in the relevant sections of the paper to avoid any ambiguity.
>    3. Correct $R$ to $\mathbb{R}$.

---

> ### Author Response · Authors · 2024-11-23
>
> **To C3:** In team adversarial games, TMECom is NE discussed in Section 2. We have provided explanations in the relevant sections of the paper to prevent any misunderstandings for the readers.
>
> **To C4:** Our platform is designed to be format-agnostic when it comes to graph data input. As long as users can obtain the adjacency list representation of their graph, regardless of the original format (e.g., GraphML, XML, or other standard graph formats), they can easily integrate it into our platform. Specifically, users only need to pass the adjacency list as parameters to our platform's game generation function, enabling them to customize their own games. To improve the clarity of our paper, we added a detailed explanation of supported data formats in Section 3.1.
>
> **To C5:** Our platform incorporates several technical enhancements that contribute to its faster performance in terms of the wall-clock time. First, we have adopted the Gymnasium for game simulation, replacing the custom class implementations found in the original papers. This change results in faster simulation processes and eliminates redundant data copying operations, leading to improved efficiency.
>
> Additionally, we have implemented various code optimizations to enhance the platform's performance. These include improved data type conversions, such as using numpy-to-tensor conversions instead of list-to-tensor operations, which reduces processing time. We have also focused on enhancing memory management throughout the platform, resulting in more efficient resource utilization.
>
> To address the readers' concern about performance improvements, we added a detailed discussion in Appendix E comparing GraphChase's simulation speed with original implementations.
>
> **To C6:** We agree that using "pursuer" and "evader" better reflects the abstract nature of our mathematical model and helps avoid potential misinterpretations about real-world applications.
>
> We have revised the paper as follows: 1. Use "pursuer" and "evader" as the primary terminology throughout the theoretical discussions and model descriptions; 2. Maintain references to "police officers" and "criminals" only in specific illustrative examples where concrete scenarios help readers understand the practical applications of the abstract model.

---

> > ### Comment · Reviewer_FH55 · 2024-11-26
> >
> > Thanks for engaging with my comments!
> >
> > Regarding W1, the response is essentially reiterating the flexibility of the work (and effectively answering another question). It does not address the core of the comment: the lack of experiments in a variety of scenarios. In my opinion, the authors need to demonstrate that the framework can indeed accommodate the type of experiments that are described, by putting themselves in the "shoes" of the researchers who would use this framework themselves. I am confident that, in trying this, issues, bugs, shortcomings of functionality etc. will be found. What is currently there seems more of a vision than an actual framework that is ready-to-use. It is inaccurate, in my opinion, to describe it as such without demonstrating a wide range of experiments.
> >
> > I am therefore keeping the original recommendation: I strongly believe the "framework" claims are not supported without demonstrating it can be used as such. The hard work to put together the problems and implementations has already be done. I would encourage the authors to revise the paper to include such experiments and resubmit to another venue in the future. The reviews should provide encouragement that this is worth pursuing, but the work is not ready for publication at this time in my opinion.

---

> > > ### Author Response · Authors · 2024-12-01
> > >
> > > Thank you for your previous response. We would appreciate your feedback on whether our additional response has adequately addressed your concerns. If there are any remaining questions or points that need further clarification, we would be happy to provide additional information.

---

> ### Author Response · Authors · 2024-11-28
>
> Thank you for your valuable feedback!
>
> We have expanded our experimental validation by incorporating a variety of scenarios used in the UNSG domain (Xue et al. 2021; 2022; Li et al. 2023a; 2024) in Section 4.2 (details are in Appendix D), which include larger $15\times 15$ grid structures and real-world maps based on Singapore and Manhattan. The $15\times 15$ grid network represents the randomly generated network, and two real-world networks represent different topological structures in real-world cities.  To our knowledge, the Singapore and Manhattan maps represent the most realistic and complex graph structures currently used in the UNSG algorithm research. We believe that our extensive testing on Singapore and Manhattan maps demonstrates GraphChase's capability to handle real-world problems.
>
> Regarding your point about testing communication between agents, to the best of our knowledge, no existing learning algorithms for solving UNSGs address this specific aspect. This limits our ability to conduct comparative experiments in this variant. However,  GraphChase provides a standardized testing platform for researchers interested in exploring such scenarios. We hope that GraphChase will be a foundation to advance future UNSG research.

---

### Official Review · Reviewer_Wrzs · 2024-11-03

**Soundness:** 3
**Presentation:** 3
**Contribution:** 3
**Rating:** 6
**Confidence:** 4

**Summary:**

This is a benchmarks paper and the authors have chosen the correct primary area for this work.
The work proposes an environment (or simulator) for urban security game, with various parameters that can be tuned for changing game parameters. This is a well-studied multi-player (>2) problem. The authors also evaluate many known algorithms in this environment.

**Strengths:**

1) The work is an important step towards benchmarking performance of the different algorithms and evaluating them on the same setup.
2) Benchmarking in multi-agent systems is lacking, so this is a timely piece of work.

**Weaknesses:**

1) There should a list/table of parameters that can be controlled. Right now, it is all in text and getting missed.
2) The paper mentioned NE many times, but the games seem stochastic form, and probably extensive form. There have Markov equilibrium and Subgame perfect equilibrium. It is not clear to the reviewer if the framework can handle all of these - as this is just a simulator, it is probably ok.
3) Any simulator must deal with question of realism. The authors acknowledge the abstract nature of the simulator model itself. But, then the important question is if the simulator is extendible easily to handle other aspects if some researcher wants to add extra nuances of the real world. How is this handled, please explain?
4) The comparison to other works in this area---in the design of environment and simulator---must be expanded. What extra does this work offer over SIMPE and Avalon?

**Questions:**

Please respond to questions in weakness. Note that while I am positive, my overall view is still dependent on the questions I raised, so please answer these in details.

---

> ### Author Response · Authors · 2024-11-23
>
> We thank the reviewer for constructive advice.
>
> ## Weaknesses
>
> 1. **To Weakness 1**: We added user-controllable parameters in Appendix B.
>
> 2. **To Weakness 2**: As you correctly mentioned, our platform functions primarily as a simulator. While our current testing algorithms focus on Nash Equilibrium solutions, it's important to clarify that the platform itself is equilibrium-concept agnostic.
>
>    The platform serves as a testing and training environment, providing researchers for algorithm development and evaluation. Researchers interested in studying Markov equilibrium or subgame perfect equilibrium would need to implement their own solution algorithms. The platform's role is to provide the environmental framework within which these various equilibrium concepts can be explored, rather than being tied to any specific equilibrium solution concept.
>
>    One of our platform's aims is to allow researchers to implement and test different solution concepts according to their research needs. So the simulator's architecture is intentionally separated from the specifics of equilibrium computation, making it a versatile tool for studying various game-theoretic solution concepts.
>
> 3. **To Weakness 3**: Our platform is designed with extensibility as a core feature, allowing researchers to incorporate additional real-world elements into their models easily. The key to this flexibility lies in the method of generating games. Researchers can design and input their own adjacency list as parameters to our platform's game generation function, enabling customized environment modeling for specific research requirements.
>
>    What's more, the graph implementation is based on the NetworkX library, so our platform enables researchers to incorporate diverse graph attributes, such as edge-specific travel times and other details to model real-world scenarios.
>
>    The platform's modular design ensures that such modifications can be implemented without requiring changes to the core system architecture.
>
> 4. **To Weakness 4**: SIMPE is a Matlab-based platform for simulating Pursuit-Evasion games which has several limitations. First, it outputs the coordinates of the pursuer and evader in the x-y plane, with a continuous position space. So it is difficult to model the topological structure of actual UNSGs. Secondly, it does not take time information into account, overlooking the temporal constraints inherent in UNSG problems. Finally, it offers only three predefined strategies for pursuers and evaders, which substantially restricts the platform's extensibility.
>
>    Avalon, on the other hand, is designed to simulate biological survival skills (from basic actions like eating to complex behaviors like hunting and navigation). It provides diverse procedural 3D environments where agents must survive which is not suitable for model UNSGs. We expanded our comparison of GraphChase with SIMPE and Avalon in the related work section.
>
>    Our GraphChase platform is purposefully designed to model and simulate real-world criminal prevention scenarios, enabling the application of developed algorithms to urban security challenges. We deliberately choose Gymnasium to implement our platform, as many DRL researchers are already familiar with this environment, thus reducing the learning time. Additionally, our implementation of Gymnasium's SyncVector functions introduces parallel simulation capabilities, while Gymnasium's render functions enable researchers to better understand their algorithms' real-world performance. These features are not present in the other platforms. We hope that GraphChase can be a valuable tool for researchers to design their own algorithms.

---

> > ### Comment · Reviewer_Wrzs · 2024-11-24
> > **Thank you**
> >
> > I am happy with the author responses.

---

> > > ### Author Response · Authors · 2024-11-30
> > >
> > > Thank you once again for your positive review.

---

### Author Response · Authors · 2024-11-23

We sincerely appreciate the reviewers' constructive feedback. We believe we have addressed all the reviewers' concerns. In response to their comments, we have made several supplements to our manuscript:

- In Appendix C, we added a parameter table that users can control to generate graph structures tailored to their specific research needs.
- Now we have additional experiments on a variety of scenarios used in the UNSG domain (Xue et al. 2021; 2022; Li et al. 2023a; 2024) in Section 4.2 (details are in Appendix D), which includes larger $15\times 15$ grid structures and real-world maps based on Singapore and Manhattan. We believe that our extensive testing on these maps effectively demonstrates GraphChase's capability to address real-world problems.
- Regarding the convergence speed, we provided a detailed comparison in Appendix E to explain why GraphChase achieves faster convergence compared to the original papers in terms of the wall-clock time.
- We have also included a brief tutorial on implementing GraphChase. The details are shown in Appendix F.

We have made some revisions, which are highlighted in red in our updated manuscript. We sincerely thank the reviewers for their valuable feedback. We welcome further discussion and are open to any questions or suggestions you may have.

---

### Meta-Review · Area_Chair_mZZY · 2024-12-20

**Metareview:**

The paper introduces GraphChase, an open-source platform for Urban Network Security Games (UNSGs). UNSGs model complex scenarios involving allocating limited security resources in urban environments and balancing cooperative and adversarial interactions between multiple agents. The authors aim to provide a unified platform to support researchers in developing, testing, and benchmarking algorithms for UNSGs. Experimental results demonstrate that GraphChase improves computational efficiency compared to baseline implementations and supports a wide range of game configurations.

Reviewers appreciated the platform's potential to address a significant gap in UNSG research by offering a standardized environment for algorithm development and evaluation. However, they expressed concerns about the paper's readiness for publication due to limitations in experiments, insufficient clarity in contributions, and the need for a broader evaluation of the platform's versatility.

Reviewer HG6y emphasized the need for more systematic experiments in complex environments. Reviewer eAes appreciated GraphChase but pointed out the limited scope of experiments and unclear presentation of related work. Reviewer FH55 raised similar concerns about the lack of experimental depth and noted that the current presentation sometimes reads more as an advertisement than objective scientific writing. Reviewer Wrzs found the platform's extensibility promising but suggested further comparisons with related works and additional implementation details to strengthen the contribution.

During the discussion, the authors provided additional experiments on real-world graph structures, including Singapore and Manhattan maps, and clarified several technical details. These revisions partially addressed the reviewers' concerns, but a consensus emerged that the paper requires further polishing. Specifically, the reviewers agreed that demonstrating broader experiments, improving clarity, and conducting more systematic evaluations of GraphChase’s capabilities would significantly strengthen the work.

Given these considerations, the reviewers recommend a rejection at this stage. However, they acknowledge the importance of the problem and the potential of the proposed platform. The authors are encouraged to address the concerns raised and resubmit the paper to a future venue.

**Additional Comments On Reviewer Discussion:**

The authors and the Reviewers engaged in productive discussion, which led to improvements to the manuscript.

Reviewers agreed that more comprehensive experiments, particularly in complex and realistic environments, are necessary to validate the platform's utility. The authors responded with additional experiments on the Singapore and Manhattan maps. Reviewer HG6y acknowledged these efforts, stating, “The new experiments in the Singapore map...strengthen the paper,” but insisted that “more systematic experiments in more complex environments...and a more careful presentation perspective to highlight the key contributions of the platform...are required.” Similarly, reviewer eAes appreciated the new details. Still, they emphasized that the platform's potential to “support the implementation of various evaluation methods for pursuers' strategies, as well as the computation and assessment of the overall equilibrium” could have been further substantiated.

Finally, Reviewer Wrzs highlighted the need for detailed comparisons with existing platforms like SIMPE and Avalon ("The comparison to other works in this area...must be expanded. What extra does this work offer over SIMPE and Avalon?”).

---

### Decision · Program_Chairs · 2025-01-22

Reject